# Artificial Intelligence: A Promising Tool for Application in Phytopathology

Victoria E. González-Rodríguez [ID], Inmaculada Izquierdo-Bueno [ID], Jesús M. Cantoral [ID], María Carbú *[ID] and Carlos Garrido *[ID]

Laboratorio de Microbiología, Departamento de Biomedicina, Biotecnología y Salud Pública, Facultad de Ciencias del Mar y Ambientales, Universidad de Cádiz, 11510 Puerto Real, Spain; victoriaeugenia.gonzalez@uca.es (V.E.G.-R.); inmaculada.izquierdo@uca.es (I.I.-B.); jesusmanuel.cantoral@uca.es (J.M.C.)
* Correspondence: maria.carbu@uca.es (M.C.); carlos.garrido@uca.es (C.G.)

**Abstract:** Artificial intelligence (AI) is revolutionizing approaches in plant disease management and phytopathological research. This review analyzes current applications and future directions of AI in addressing evolving agricultural challenges. Plant diseases annually cause 10–16% yield losses in major crops, prompting urgent innovations. Artificial intelligence (AI) shows an aptitude for automated disease detection and diagnosis utilizing image recognition techniques, with reported accuracies exceeding 95% and surpassing human visual assessment. Forecasting models integrating weather, soil, and crop data enable preemptive interventions by predicting spatial-temporal outbreak risks weeks in advance at 81–95% precision, minimizing pesticide usage. Precision agriculture powered by AI optimizes data-driven, tailored crop protection strategies boosting resilience. Real-time monitoring leveraging AI discerns pre-symptomatic anomalies from plant and environmental data for early alerts. These applications highlight AI's proficiency in illuminating opaque disease patterns within increasingly complex agricultural data. Machine learning techniques overcome human cognitive constraints by discovering multivariate correlations unnoticed before. AI is poised to transform in-field decision-making around disease prevention and precision management. Overall, AI constitutes a strategic innovation pathway to strengthen ecological plant health management amidst climate change, globalization, and agricultural intensification pressures. With prudent and ethical implementation, AI-enabled tools promise to enable next-generation phytopathology, enhancing crop resilience worldwide.

**Keywords:** artificial intelligence; phytopathology; emerging disease; climate change; control diseases

## 1. Introduction

Plant diseases have plagued agricultural crops for centuries, presenting a persistent threat to global food security [1,2]. Annually, plant diseases account for an estimated 10–16% of global crop losses, translating into profound economic impacts [3,4]. With the global population projected to reach 9.8 billion by 2050, it is imperative to increase crop yields by 25–70% to meet escalating food demands [5], emphasizing the need for revolutionary advancements in managing plant diseases.

However, the dynamics of plant pathosystems are complex, influenced by genetic and environmental factors, and challenged by the evolution of host–pathogen interactions [6,7]. These interactions have been significantly altered in recent decades due to anthropogenic factors, particularly climate change and modern agricultural practices. Climate change has been a critical driver in the emergence and spread of new plant pathogens, altering the geographical distribution of existing diseases and creating favorable conditions for the emergence of novel pathogens [8,9]. Moreover, the intensification of agricultural practices, including the use of monocultures and high-input farming systems, has reduced crop diversity, making them more susceptible to widespread disease outbreaks [10–12].

These evolving dynamics necessitate innovative solutions to expedite the discovery of knowledge in plant disease dynamics, enhance crop resilience, and understand plant–microbe interactions. Artificial intelligence (AI) offers groundbreaking avenues for deciphering the complexity of plant pathosystems and deriving practical insights for disease management [13,14]. The capacity of AI to analyze large volumes of agricultural data enables the revelation of correlations beyond human cognitive abilities [15,16]. This capability positions AI as a formidable tool in more easily unravelling the nature of plant–disease interactions. These studies involve a substantial amount of data, and AI can identify behavioral patterns in ways that are not readily discernible through purely human analysis. Furthermore, machine learning algorithms can continually self-improve, progressively facilitating the interpretation of new datasets in similar studies, while discarding data that pertain to the inherent variability of the studies [17,18].

This review has three central aims: (1) examine existing and emerging applications of AI supporting plant disease management; (2) identify current challenges and gaps hindering the adoption of AI-driven solutions; and (3) outline a roadmap for stakeholder alignment to mainstream AI in crop protection practices. By realizing these objectives through a detailed literature analysis, this review seeks to catalyze a strategic transition toward AI-enabled plant disease science and agriculture worldwide as a bridge to more sustainable food production, addressing these evolving challenges in plant pathosystems.

## 2. Overview of Phytopathology

Phytopathology is the scientific discipline dedicated to the study of plant diseases. This field investigates the complex interactions between plants and pathogenic organisms, shedding light on the mechanisms underlying the onset and progression of diseases. The scope of phytopathology encompasses the etiology of diseases, their epidemiology, and the development of integrated strategies for managing them in agricultural and horticultural contexts [2]. It is estimated that over 50,000 species of plant pathogens cause damage to more than 30,000 plant species [1]. These pathogens comprise various taxa, including fungi, bacteria, viruses, viroids, protozoa, and algae. Each pathogen type prompts unique disease manifestations and demands tailored investigative approaches. Furthermore, the effects of climate change, globalization, and crop intensification add complexity to deciphering modern plant disease epidemiology [8].

As a discipline so integral to food security and agricultural sustainability, the importance of phytopathology cannot be overstated. As mentioned earlier, plant diseases result in substantial economic losses in major staple crops worldwide, amounting to USD 220 billion in annual economic damages globally [3,4]. For instance, Fusarium wilt disease alone results in approximately USD 410 million in annual banana crop damages, while cassava brown streak disease incited over USD 100 million in crop damages across eastern Africa in the early 1990s [19,20]. By elucidating plant–pathogen interactions and disease epidemiology, phytopathology enables breeding disease-resistant varieties, optimizing cultural practices, and implementing integrated pest management interventions that minimize disease impacts and crop loss [6]. The development of resistant cultivars alone has saved certain crops from near extinction, as exemplified by saving papaya production in Hawaii from papaya ringspot virus in the mid-20th century [21]. A recent example of success in phytopathology is the management of coffee rust disease in Central America. Since 2012, coffee rust has significantly threatened coffee production, but the implementation of resistant varieties and improved agronomic practices has resulted in a notable recovery in affected regions [22]. Another case is the management of citrus tristeza virus in Florida, where the use of tolerant rootstocks and vector control has helped mitigate the impacts of the disease [23].

However, current disease management strategies often provide incomplete and temporary solutions in the face of an evolving pathogen landscape. In Figure 1, we present a conceptual framework that lists some of the major challenges in contemporary phytopathology, including emerging diseases, climate change, global trade and pathogen dissemination,

breakdown of resistance, and data analysis and integration. The examples cited illustrate how phytopathological science must respond to specific diseases with innovations and adaptive strategies, highlighting its relevance in an ever-changing agricultural world.

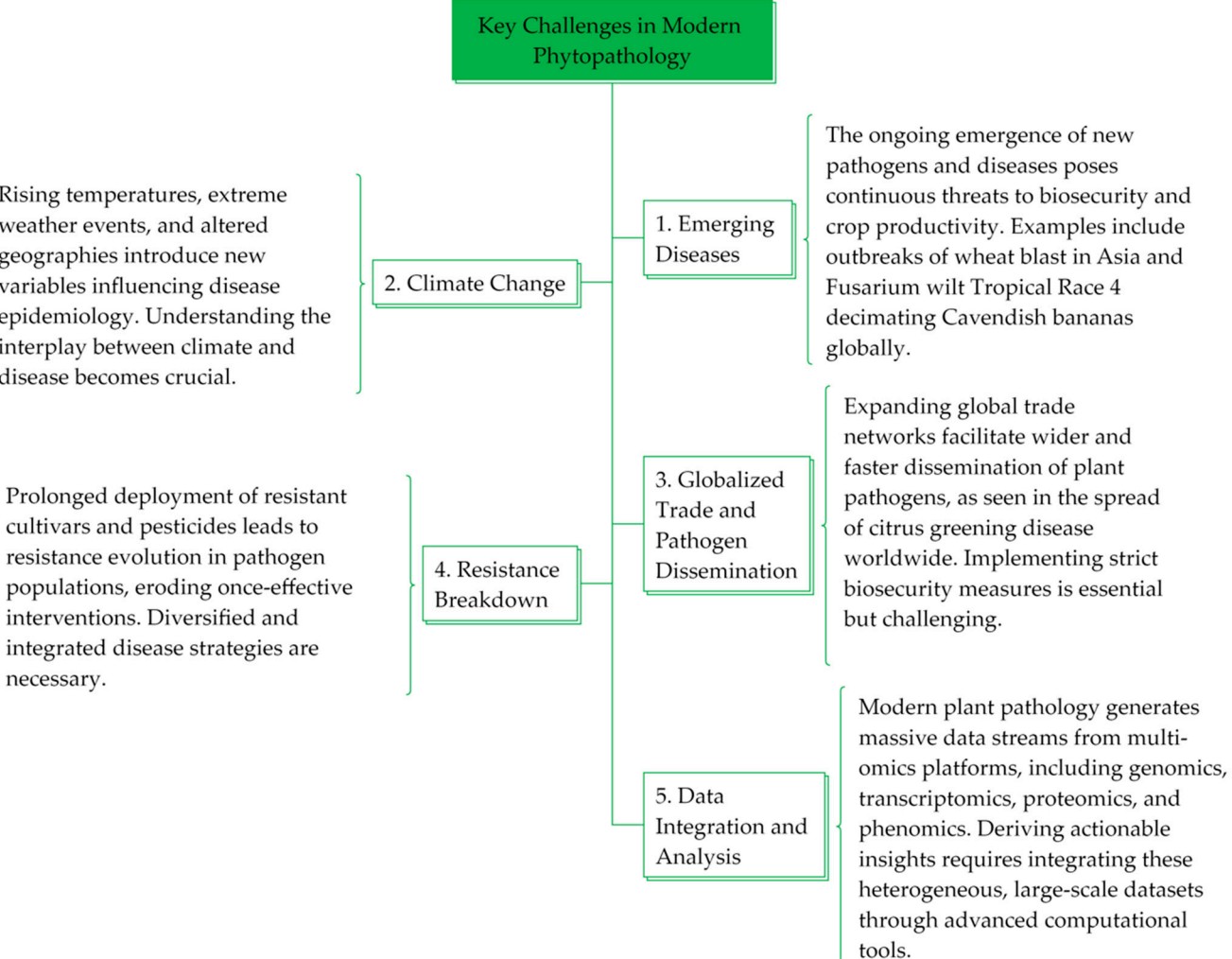

**Figure 1.** Several key challenges for innovation in modern phytopathology.

## 3. Role of Technology in Phytopathology

Historically, phytopathologists predominantly relied on conventional methods, such as visual inspection, symptomatology characterization, and pathogen isolation for plant disease diagnosis and management [2]. While these traditional techniques are valuable, they have inherent limitations, especially when considering the emerging agricultural challenges of the modern world. For instance, visual disease symptoms often do not manifest until infections are well-established, leading to delayed intervention and unchecked pathogen spread [24]. Reliance on visual symptoms alone also poses challenges in distinguishing between diseases with similar outward manifestations [25].

Traditional methods, such as pathogen isolation and culture, remain cornerstones in diagnostics. They require time-consuming processes, and obtaining pure cultures can be technically challenging [26]. Furthermore, many phytopathogenic microbes exhibit complex life cycles, switching between morphological forms, which traditional techniques often fail to detect at low pathogen levels or in identifying novel strains [20,27]. This limits their reliability and applicability in the dynamic agricultural ecosystems of today.

*3.1. Advent of Emerging Technologies in Agriculture*

The advent of emerging technologies and advanced analytical tools has significantly altered the agricultural landscape. Next-generation high-throughput DNA sequencing platforms, for instance, have revolutionized plant–microbiome studies, enabling the rapid genomic characterization of plant-associated microbiota and pathogens [28,29]. Metagenomic approaches have elucidated complex plant–microbe interactions, identified novel pathogens, and assessed microbiome shifts correlating with health–disease transitions. Additionally, ultra-sensitive quantitative DNA and RNA diagnostic tests now facilitate the detection of exceedingly low pathogen levels at early infection stages [25,30].

Remote sensing technologies and high-resolution spectral imaging through satellites, planes, and unmanned aerial vehicles offer large-scale capabilities in monitoring crop health and stress levels [31,32]. These tools enable the real-time, non-invasive assessment of plant vigor and the detection of disease outbreak locations in the field, facilitating timely and precise management interventions [33]. Recent advancements in nano-biosensors and lab-on-chip devices have allowed for the continuous monitoring of environmental parameters influencing disease development, such as temperature, humidity, soil water content, and microclimate conditions [34]. The integration of these sensors in agricultural ecosystems generates comprehensive datasets, shedding light on the crop–climate–disease interplay [35].

Big data analytics, automation, robotics, and artificial intelligence (AI) are accelerating a paradigm shift towards data-driven precision agriculture systems [36–38]. Phytopathology, transitioning into a highly interdisciplinary and technology-intensive science, integrates diverse data streams. Advanced computational methods offer immense promise in deriving actionable insights from the wealth of agricultural big data for efficient disease management [39].

*3.2. Need for Advanced Data-Driven Solutions*

While emerging technologies provide promising avenues, significant challenges persist in effectively managing diseases within the highly complex and dynamic agricultural ecosystems of today. Globalization, climate change, and intensive farming systems facilitate the increased emergence and faster evolution of plant pathogens [8,40]. Many conventional disease management approaches now face diminishing effectiveness due to rising pathogen resistance, alongside serious environmental and health concerns [11,12,41].

The complexity characterizing plant–pathogen interactions and disease epidemiology necessitates a paradigm shift towards sophisticated, integrated solutions. In this context, AI and advanced machine learning algorithms emerge as potentially transformative tools in modern data-driven phytopathology. Machine learning models can analyze vast, disparate datasets, including weather, soil, plant omics, microbiome, and pathogen genomic information [42]. These models discern subtle multivariate relationships, predict disease outbreak risks, and enable targeted intervention strategies undetectable via conventional approaches [13,14,42]. Continually learning from accumulating agricultural data streams, such AI-based systems progressively improve their predictive capabilities and decision support functionalities. Therefore, harnessing modern technology and computational innovation is imperative for developing dynamic, ecologically balanced, and economically viable plant disease management regimes, crucial in addressing the pressing food security challenges of the future [43].

## 4. Introduction to Artificial Intelligence (AI)

Artificial intelligence (AI) represents a transformative paradigm in computing, revolutionizing how machines perform tasks that typically require human intelligence [44,45]. In this section, we delve into the fundamental aspects of AI, tracing its evolution, understanding its basic principles, and exploring its relevance to the field of phytopathology.

### 4.1. Definition and Basics of AI

Artificial intelligence (AI) is the capacity of computer systems to undertake tasks that usually require human cognition, such as learning, reasoning, perception, prediction, and decision-making [46]. At its foundation, AI is about developing algorithms allowing machines to emulate aspects of human intellect, like processing and adapting to information over time. AI spans several sub-domains: machine learning uncovers data patterns without direct programming [47], computer vision gives machines sight [48], and natural language processing enables them to understand and produce human language [49]. Expert systems represent human knowledge in structured domains [50], and robotics combines these AI capabilities for environmental interaction [51–53]. AI can be general, with human-like cognitive abilities, or narrow, focused on specific tasks, where most advancements occur, revolutionizing industries with applications like smart assistants and fraud detection [54,55]. Deep learning, a machine learning subset using multi-layer neural networks, exemplifies AI's core research areas, alongside computer vision and natural language processing, pushing autonomous pattern recognition and decision-making further [47]. Table 1 presents different AI models, detailing their definitions and pivotal development dates, illuminating AI's diverse landscape. Furthermore, the integration of deep learning algorithms optimizes neural network performance, as highlighted in [56], underscoring ongoing innovations in AI's algorithmic framework.

**Table 1.** Different artificial intelligence models. Definition and significant dates of development are included.

| Artificial Intelligence Models | Definition and Significant Dates | Reference |
| --- | --- | --- |
| LLM—Large Language Model | These are systems that use large-scale neural networks to understand and generate human-like language. They excel in natural language processing tasks, such as text completion and language translation. Notable developments in large language models, especially the introduction of GPT-3, occurred around 2020–2021. | [57] |
| CNN—Convolutional Neural Network | A type of neural network designed for image processing and recognition. It uses convolutional layers to automatically and adaptively learn spatial hierarchies of features from input images. Proposed by Yann LeCun in the early 1990s, CNNs gained prominence in the mid-2010s with breakthroughs in image recognition tasks. | [58,59] |
| RNN—Recurrent Neural Network | A type of neural network architecture designed to recognize patterns in sequences of data. RNNs are well suited for tasks involving sequential data, such as time series analysis and natural language processing. While the concept of RNNs dates back to the 1980s, their resurgence and success in various applications, especially in natural language processing, gained momentum in the mid-2010s. | [60] |
| GAN—Generative Adversarial Network | GANs consist of two neural networks, a generator and a discriminator, which are trained simultaneously through adversarial training. GANs are used for generating new, realistic data instances, such as images. Introduced by Ian Goodfellow and his colleagues in 2014, GANs have since become a revolutionary concept in the generation of realistic data. | [61] |
| Decision Tree and XGBoost (eXtreme Gradient Boosting) | They are powerful models for classification, regression, and ranking tasks. Decision Trees are simple yet effective models that partition data into subsets based on feature values, using a tree-like structure of decisions and their possible consequences. XGBoost, an implementation of gradient boosted decision trees designed for speed and performance, significantly improves model accuracy by combining multiple decision trees to correct the errors of predecessors. Introduced by Chen and Guestrin in 2016, XGBoost has become a dominant force in machine learning competitions due to its efficiency and effectiveness. | [62,63] |

| Artificial Intelligence Models | Definition and Significant Dates | Reference |
|---|---|---|
| ElasticNet, Lasso, and Ridge Regression | They are regularization techniques in linear regression that address overfitting by penalizing the size of coefficients. ElasticNet combines the properties of both Lasso (Least Absolute Shrinkage and Selection Operator) and Ridge Regression by integrating their penalty terms; it is particularly effective when dealing with highly correlated data. Lasso contributes to feature selection by reducing the coefficients of less important features to zero, while Ridge Regression shrinks the coefficients but does not set them to zero. <br><br> These methods were developed in the early 21st century, with ElasticNet introduced by Zou and Hastie in 2005, offering a bridge between Lasso's feature selection and Ridge's coefficient shrinkage. | [64] |
| Random Forest | It is an ensemble learning method renowned for its versatility and accuracy in classification and regression tasks. By constructing multiple decision trees at training time and outputting the mode of the classes (for classification) or mean prediction (for regression) of the individual trees, Random Forest mitigates the overfitting problem common to single decision trees. <br><br> This model's significant development dates back to the early 2000s, with Breiman's seminal paper in 2001 laying the foundational framework for Random Forest algorithms. | [65] |
| SVM—Support Vector Machine | A supervised machine learning algorithm used for classification and regression analysis. SVMs are effective in high-dimensional spaces and are particularly useful in tasks like image classification and handwriting recognition. <br><br> Proposed by Vladimir Vapnik and Corinna Cortes in the 1990s, SVMs gained popularity in the early 2000s and became a staple in machine learning applications. | [66,67] |
| KNN—k-Nearest Neighbors | A simple and effective algorithm used for classification and regression tasks. KNN makes predictions based on the majority class or average of the k-nearest data points in the feature space. <br><br> KNN is a classical algorithm, and its principles have been known for decades. It is widely applied in various fields since the 1960s. | [68,69] |
| DNN—Deep Neural Network | A neural network with three or more layers, including an input layer, one or more hidden layers, and an output layer. Deep neural networks are capable of learning intricate representations and are used in various applications. <br><br> While the concept of deep neural networks has roots in the 1960s, their resurgence and practical success came in the mid to late 2000s with advancements in training algorithms and hardware. | [70] |
| MLP—Multilayer Perceptron | MLP is an artificial neural network model consisting of an input layer, multiple hidden layers, and an output layer, with each layer fully connected to the next. It employs backpropagation for learning, allowing it to model complex non-linear relationships. <br><br> Developed in the 1980s, MLPs are versatile in applications ranging from pattern recognition to classification and regression tasks, marking a significant advance in the field of deep learning. | [71] |
| SGD—Stochastic Gradient Descent | It is an optimization algorithm pivotal for training a broad spectrum of artificial intelligence models, notably in deep learning. It optimizes model parameters by calculating gradients based on randomly selected data subsets, enhancing training efficiency across large datasets. Introduced in the context of machine learning in the late 20th century, its conceptual roots trace back to Robbins and Monro's stochastic approximation method in 1951, laying the theoretical groundwork for iterative stochastic optimization techniques in AI. | [72] |

**Table 1.** *Cont.*

| Artificial Intelligence Models | Definition and Significant Dates | Reference |
|---|---|---|
| LSTM—Long Short-Term Memory | A type of recurrent neural network architecture designed to overcome the limitations of traditional RNNs in capturing long-term dependencies in sequential data. LSTMs are widely used in natural language processing and speech recognition. Proposed by Sepp Hochreiter and Jürgen Schmidhuber in 1997, LSTMs became popular in the mid-2010s, addressing challenges in capturing long-term dependencies. | [60,73] |
| RL—Reinforcement Learning | An area of machine learning where an agent learns to make decisions by interacting with an environment. The agent receives feedback in the form of rewards or penalties, allowing it to learn optimal strategies over time. RL has a history dating back to the 1950s and 1960s, but recent advancements, especially in deep reinforcement learning, have gained prominence since the mid-2010s. | [74,75] |
| BERT—Bidirectional Encoder Representations from Transformers | A pre-trained natural language processing model based on transformer architecture. BERT is particularly effective in understanding the context of words in a sentence and is used for various language-related tasks. Introduced by Google AI in 2018, BERT brought a breakthrough in natural language processing by capturing contextual information bidirectionally. | [76] |

*4.2. Evolution of AI*

The evolution of AI began with early conceptualizations by figures like Ada Lovelace and Alan Turing, progressing through the 1950s with attempts at creating intelligent machines [46] (Figure 2). Despite initial successes, challenges led to periods of stagnation, known as "AI winters" [77]. The 21st century marked a resurgence, fueled by advancements in computational power, data generation, and machine learning, leading to breakthroughs in areas like vision and speech [47]. Recent developments in natural language processing indicate significant advancements in AI's ability to understand and generate human-like language [78], hinting at future communicative AI systems' potential [79–81] and supported by large-scale data and computational resources [82]. Rapid innovation continues toward safer and more robust language models aligned with human values [83]. The historical context of AI's development underscored by data-driven approaches is detailed in Figure 2.

*4.3. Relevance of AI in Various Fields*

In the realm of artificial intelligence (AI), its transformative impact extends far beyond theoretical frameworks, finding tangible applications in diverse fields. AI is making significant strides, revolutionizing industries and scientific endeavors.

4.3.1. Healthcare: Enhanced Diagnostics and Personalized Medicine

In the healthcare sector, AI is significantly improving clinical diagnostics and personalizing medicine. It utilizes advanced algorithms for medical image analysis and genomic pattern recognition, enabling the early detection of conditions not visible through conventional methods [84]. In digital pathology, AI predicts the risk of cancer metastasis earlier than traditional clinical indicators [85–87]. For drug discovery, deep learning has shortened the timeline for developing new medications [88]. AI also customizes treatment plans to individual genetic profiles, enhancing patient outcomes [89]. This integration of AI is transforming healthcare diagnostics, treatments, and drug development processes.

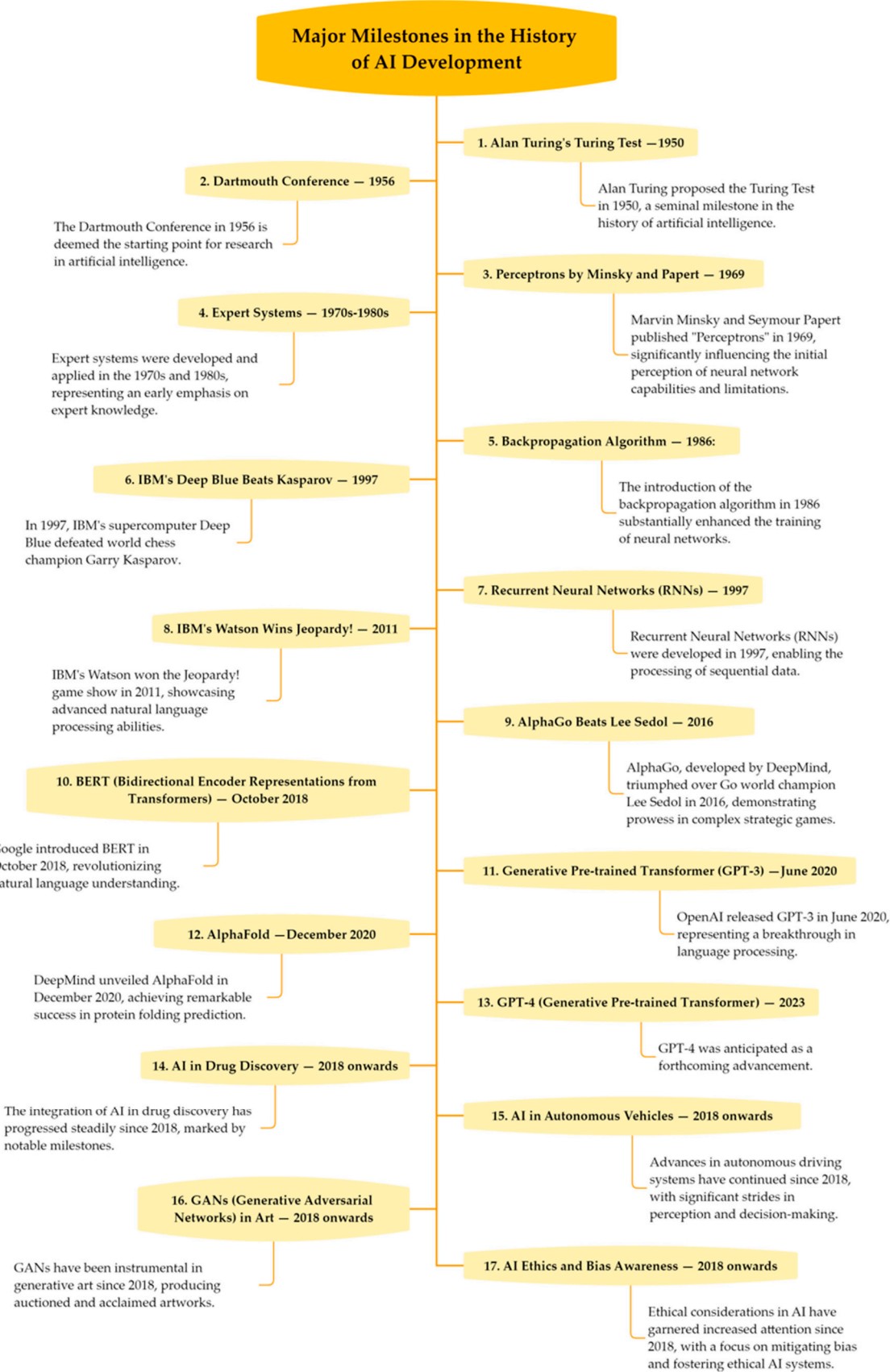

**Figure 2.** Major milestones in the history of AI development.

### 4.3.2. Engineering: Optimizing Complex Systems

In the field of engineering, AI plays a crucial role in optimizing complex systems through autonomous design enhancements, predictive maintenance, and self-adaptive mechanisms. It has made significant strides in improving aerodynamic designs, reducing the development time for aircraft wing prototypes [90], and employing AI vision systems for the early detection of structural weaknesses, such as bridge cracks, to facilitate timely repairs [91]. Moreover, neural networks allow electronics to dynamically reconfigure circuitry, enhancing resilience and operational efficiency despite component damages [92,93], underscoring AI's profound impact on fostering innovation and efficiency in engineering.

### 4.3.3. Business: Data-Driven Decision Automation

In the business sector, AI is revolutionizing decision-making by automating complex analyses previously dependent on human judgment. It leverages advanced algorithms to process diverse datasets, from historical records to market trends, enhancing risk management and optimizing processes [94]. In finance, AI, through deep reinforcement learning, outperforms human stock trading strategies [95]. Additionally, AI rapidly analyzes complex legal documents and drives business analytics and marketing decisions, offering strategies that significantly improve efficiency and outcomes [96,97]. This marks AI's significant role in advancing business practices through data-driven insights.

### 4.3.4. Transportation: Optimized Mobility

In transportation, AI enhances urban mobility and develops autonomous driving technologies. Adaptive traffic systems, using deep learning, coordinate signals to reduce congestion, significantly improving travel times [98]. Autonomous vehicles combine sensor data for real-time environmental perception, crucial for safe navigation, including cameras, LiDAR, radar, GPS, wheel odometry, and IMUs [99,100]. The integration of sensors, software, and AI computational power is indeed enhancing the safety of autonomous vehicles [101]. For instance, Waymo's autonomous cars have covered over 20 million miles, demonstrating sophisticated navigation capabilities [102]. Recent studies focus on integrating AI into automotive manufacturing and navigation systems, promising safer, more efficient transportation solutions [103,104].

### 4.3.5. Space Exploration: Autonomous Exploration and Data Analysis

AI is transforming space exploration by enabling autonomous operations and advanced data analysis. Machine learning equips spacecraft and rovers, like NASA's Mars rovers, to navigate and collect data independently, optimizing paths in real time. Research expands AI's role in autonomous spacecraft technology and space law [105,106], with a focus on developing trusted AI systems for mission autonomy [107]. These AI advancements promise to significantly enhance the efficiency, autonomy, and intelligence of space missions, marking a new era in space exploration and analysis.

### 4.3.6. Education: Personalized Learning and Student Support

AI is revolutionizing education sector by facilitating personalized learning and supporting students. It employs algorithms in adaptive platforms to customize educational content for individual learners, enhancing their educational journey. AI-driven chatbots offer instant assistance and guidance. For example, Duolingo uses AI to adjust language lessons based on user progress [108]. Research in this area includes analyzing AI's impact on personalized learning [109], studying AI's role in learning methodologies [110], and employing machine learning to identify learning styles [111], illustrating AI's transformative potential in education.

## 5. Applications of AI in Phytopathology

Artificial intelligence (AI) is transforming approaches in phytopathology, catalyzing innovations in understanding, managing, and mitigating plant diseases. AI's capacity to

analyze vast datasets reveals subtle correlations in plant–pathogen interactions, granting key insights for disease control [16]. This section surveys prominent applications of AI across major facets of phytopathology.

*5.1. Disease Detection and Diagnosis*

Artificial intelligence (AI) enables rapid and precise disease detection and diagnosis, overcoming the limitations of techniques reliant on visual inspection. Numerous studies demonstrate the efficacy of AI in accurately diagnosing complex diseases. In an early example, Ramcharan et al. [14] applied deep learning techniques for detecting and diagnosing cassava diseases through image analysis. Using a convolutional neural network, they achieved diagnostic accuracy above 90%, demonstrating deep learning's effectiveness in identifying various cassava diseases. This approach not only surpassed traditional methods in terms of accuracy and speed but also enabled the implementation of these models on mobile devices, facilitating diagnosis in the field.

Similarly, Fuentes et al. [112] implemented three artificial intelligence architectures—Faster R-CNN, SSD, and R-FCN—to detect and diagnose diseases and pests in tomato plants. These architectures fall within the broader context of convolutional neural networks (CNNs), which are particularly suited for image recognition tasks due to their ability to learn spatial hierarchies of features from input images (see information in Table 1). The application of these models in the study marked a significant advancement in the application of CNNs in image recognition tasks since their proposal in the 1990s. The authors of [112] used images captured by cameras at various resolutions, both of healthy plants and plants with symptoms. With these images, they trained the artificial processing models, which significantly improved the accuracy in disease and pest recognition and reduced false positives during the training phase. This systematic approach allowed the AI system to effectively recognize nine different types of diseases and pests in tomato plants, demonstrating the capability of these models to handle complex environmental variables present in a plant's surroundings. Following in the footsteps of these works, but not focused on a specific plant species, Sladojevic et al. [113] also used deep convolutional neural networks (CNNs), training the artificial model with an extensive database, which allowed it to distinguish between different types of diseases in the leaves of various genera and species. The novelty and advancement of the developed model lie in its simplicity, where healthy leaves and background images are aligned with other classes, allowing the model to distinguish between diseased and healthy leaves or their surroundings using deep CNNs. The experimental results showed an accuracy of between 91% and 99% in separate class tests and an overall accuracy of 95.8% in the trained model. These studies are a clear example of CNNs' ability to handle the complexity of visual data and improve the accuracy of automated diagnosis [113].

Recent studies have continued to demonstrate AI's potential in plant disease detection and diagnosis using more modern, precise, and powerful models thanks to the development of new AI capabilities. In 2022, Arinichev [114] explored the use of artificial intelligence technologies for diagnosing fungal diseases in cereals, specifically in wheat and rice, through methods of vision and automated recognition. This analysis revealed that artificial neural networks have the capability to detect and classify disease patterns, such as yellow spots, yellow and brown rust, and brown spots, with classification metrics ranging between 0.95 and 0.99. To advance in this line of research, Arinichev examined four well-established and relatively light convolutional neural network (CNN) architectures, namely, GoogleNet, ResNet-18, SqueezeNet-1.0, and DenseNet-121, with the DenseNet-121 model particularly standing out for its optimal combination of high precision and operational efficiency. Characterized by a relatively low number of parameters and a file size suitable for mobile devices, this model achieved exceptionally high classification accuracy, surpassing the other evaluated models. Similar to previous research, such as that of Ramcharan et al., the implementation of a light neural network like DenseNet-121 facilitated its application in the field on mobile devices, allowing for quick and accurate diagnostics [114].

In the case of the study carried out by Feng et al. [115], the authors developed a convolutional neural network model for potato late blight detection method using deep learning, with high accuracy and fast inference speed, using a dataset of potato leaf disease images in single and complex backgrounds. Feng et al. used the ShuffleNetV2 2× model, characterized by its high classification accuracy, while also having a larger parameter scale and memory space compared to other models with equal accuracy. The authors improved the model through strategies that included introducing an attention module, reducing network depth, and minimizing the number of 1×1 convolutions. This resulted in an enhancement of classification accuracy while simultaneously maintaining efficient inference speed on CPUs in the devices used for its application. In the same line of work, Bracino et al. [116] carried out a study focus on the non-destructive classification of paddy rice leaf diseases using deep learning algorithms such as EfficientNet-b0, MobileNet-v2, and Places365-GoogLeNet. They aim to identify whether the rice paddy leaf is normal or infected with various diseases including bacterial leaf blight (BLB), bacterial leaf streaks (BLS), bacterial panicle blight (BPB), heart, downy mildew, hispa, or rice tungro disease (RTD). Of the models used, EfficientNet-b0 was identified as the most effective, achieving an average accuracy of 97.74%. This model is distinguished by its focus on maximizing efficiency, optimally balancing network depth, width, and the resolution of input images through a compound scaling technique, resulting in superior performance with minimal memory requirements and floating-point operations per second (FLOPS). This efficiency and precision capability distinguish it from the model used by Feng et al., the ShuffleNetV2 2×, which, although highly precise, focuses on improving inference speed and reducing parameter size through the introduction of an attention module and the optimization of the network architecture. Bracino et al.'s significant contribution lies in providing a precise and non-destructive diagnostic method for rice diseases, thereby supporting the prevention of product loss and improving crop quality through the application of advanced and efficient AI technologies.

A deep convolutional neural network model was also developed by Jouini et al. [117] to detect wheat leaf rust. The authors advanced the application of a CNN by developing a model specifically designed for the detection of wheat leaf diseases using hyperspectral images, achieving an impressive testing accuracy of 94%. This study showed the feasibility of real-time disease detection in wheat, a critical advancement for resource-constrained environments where timely and effective disease management is vital [117]. In a related study, Zhou et al. [118] introduced a novel spectral feature pseudo-graph-based residual network (SFPGRN) for the spectral analysis of plant diseases. Their method innovatively constructs a residual network model using a characteristic surface derived from natural neighborhood interpolation based on preprocessed near-infrared spectral reflection signals and first-order differential spectral index, achieving a classification accuracy of 93.21% on a dataset of apple leaf diseases and insect pests [118]. Complementing these developments, Shi et al. [93] introduced a novel fast Fourier convolutional deep neural network (FFCDNN) designed for the accurate and interpretable detection of wheat yellow rust and nitrogen deficiency from Sentinel-2 time series data. The FFCDNN model stands out for its innovative use of a fast Fourier convolutional block and a capsule feature encoder, significantly enhancing computing efficiency and model interpretability. This approach not only achieves high classification accuracy but also provides insights into the host–stress interaction, marking a significant advancement over previous studies by integrating spatial-temporal information for global feature extraction [93].

In recent times, the research group of Hassan et al. [119] introduced a groundbreaking CNN architecture for plant disease identification, leveraging inception layers and residual connections to enhance feature extraction, while employing depth wise separable convolution to significantly reduce computational complexity. This model is distinct in its ability to achieve high accuracy across various plant disease datasets with a markedly lower parameter count, illustrating a significant advancement in AI's application to phytopathology. By optimizing the model to require fewer computational resources, this work facilitates

the deployment of AI technologies on devices with limited processing capabilities, making sophisticated disease diagnosis tools more accessible to a broader range of users and applications [119].

A new evolution of CNN is the Siamese convolutional neural network (SNN). Narain et al. [120] introduced an enhanced approach to detection systems by implementing a SNN for identifying diseases in tomato leaves. Siamese neural networks stand out from conventional CNN models due to their unique structure, designed to learn to differentiate between pairs of inputs, making them exceptionally suitable for comparison and differentiation tasks. By evaluating similarities or differences between pairs of images, SNNs can offer notable accuracy in disease classification, often overcoming challenges faced by traditional CNNs in terms of intraclass variability and the scarcity of labeled data (Figure 3). In this work, Narain et al. developed a customized SNN by training with a specially collected dataset of 155 tomato leaf images, and the system demonstrated high efficacy, achieving an accuracy of 83.749% in training and 80.4% in testing. This improvement in disease classification represents a significant advancement over more classic CNN models. The implementation of Siamese networks signifies an optimization in the accuracy and efficiency of disease detection in crops, allowing for the application of appropriate management measures more quickly and accurately by providing a more robust and adaptable mechanism for recognizing complex patterns associated with various plant diseases [120].

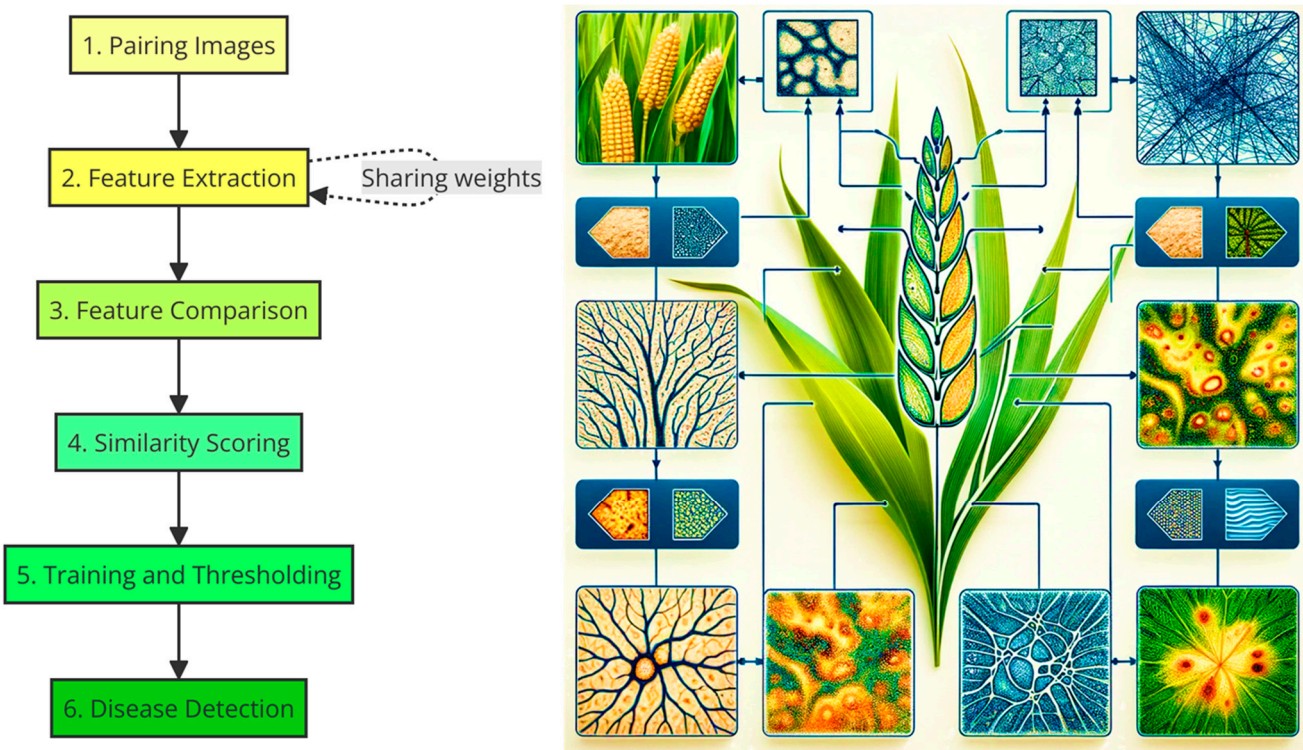

**Figure 3.** Graph diagram illustrating the operation of Siamese convolutional neural networks (SNNs) for detecting plant diseases from an image dataset.

Siamese convolutional neural networks operate into a sequence of steps illustrated in Figure 3: (a) Pairing Images: a set of image pairs of plants is created, where each pair consists of two images: it could be one of a healthy plant and one of a diseased plant, or two healthy plants, or two diseased plants. (b) Feature Extraction: each image in the pair is fed through a convolutional network that acts as a feature extractor. The key here is that both images go through the same network (sharing weights), ensuring that features are extracted uniformly. (c) Feature Comparison: the features extracted from each image are combined and fed into a layer that compares the two images. This comparison could be an absolute difference, a concatenation operation followed by dense layers, or even a

more complex metric. (d) Similarity Scoring: the network produces a score that reflects the similarity between the two images. In the context of plant disease detection, a high score might indicate that both images are of plants in the same condition (both healthy or both diseased), while a low score might suggest one is healthy and the other is diseased. (e) Training and Thresholding: during training, the network learns what features are important for distinguishing between healthy and diseased plants. A similarity threshold is adjusted that best separates pairs of healthy plant images from pairs with at least one diseased plant. (f) Disease Detection: once trained, the network can take a pair of images, process them through the network to obtain the similarity score, and using the learned threshold, determine if the plants are healthy or diseased.

Other authors are making significant advances in developing models which are more advanced in capabilities and simpler in their handling, thanks to the evolution that vision systems and their conjunction with large language modeling systems are undergoing in recent months. In this line, Tabbakh and Barpanda [121] introduced an innovative hybrid model for the classification of plant diseases, through the integration of Transfer Learning with a Vision Transformer (TLMViT). This hybrid approach stands out for its unique ability to extract and analyze deep features of plant leaf images, achieving exceptionally high accuracy in the evaluated datasets. The TLMViT is a key innovation in this study, leveraging the architecture of transformers, which has revolutionized natural language processing, to apply it in the realm of computer vision. Vision transformers adapt the concept of attention, allowing the model to focus on the most relevant parts of the image for the classification task, significantly improving accuracy and efficiency in disease identification. Tabbakh and Barpanda used a dataset freely available in the PlantVillage project, as the authors comment. This dataset encompasses more than 54,000 images of more than 38 different crop species, with a particular focus on cassava, tomato, pepper, and potato. Each image within the dataset is labeled with the plant species and, if present, the disease. This resource is freely available for computer vision and deep learning tasks, such as image classification, object detection, and semantic segmentation. In the specific research of Tabbakh et al., three different crops from the PlantVillage dataset (pepper, potato, and tomato) were used, which include 20,638 images of diseased and healthy leaves. The application of their model managed to achieve identification accuracies above 98%. This hybrid model, combining transferred learning with the power of vision transformers, illustrates a qualitative leap in the detection and classification of plant diseases, offering new perspectives for precision agriculture and sustainable crop management [121].

The integration of advanced AI models, from deep convolutional neural networks to Siamese networks and vision transformers, underscores a transformative period in the field of phytopathology. These studies collectively represent a leap forward in precision phytopathology, offering not just higher accuracy in disease diagnosis but also a model for future research to build upon. Particularly, the adoption of vision transformers marks a novel approach, leveraging the strengths of AI to address complex agricultural challenges. This evolution of AI methodologies, characterized by increased model sophistication and adaptability, promises to significantly enhance disease detection capabilities, paving the way for more targeted and effective disease management strategies.

*5.2. Advancements in Plant Disease Propagation Modeling*

The field of plant disease propagation modeling has witnessed transformative growth through the incorporation of artificial intelligence (AI) and machine learning techniques, opening new vistas in pathogen prediction and management. A pivotal approach in this field is the application of machine learning models for disease prediction based on symptoms and environmental data. In this context, the existing scientific literature encompasses a variety of meticulously developed strategies that significantly contribute to the advancement of predictive model development in the field of plant pathology.

During 2022 several studies were published to apply different algorithms and predictive models using AI. For example, a very interesting work is that published by Garrett

et al. [39] in which the authors utilized Random Forest and Support Vector Machines (SVMs) to analyze the complex interplay between climate change and pathogen emergence. Random Forest is an ensemble learning method for classification, regression, and other tasks. It operates by constructing a multitude of decision trees during training for more accurate and robust predictions (consult Table 1 and Figure 4). SVMs, in contrast, are powerful supervised machine learning models used for classification and regression challenges, effectively handling high-dimensional spaces and complex datasets. Employing these algorithms, Garrett et al. aimed to capture and model the nuanced relationships between environmental factors and the likelihood of pathogen spread, underscoring the potential of AI to offer predictive insights into plant disease dynamics influenced by climate variables. Their methodology showcases the strengths of combining multiple AI approaches to enhance predictive accuracy and provide actionable insights into pathogen management strategies [39]. These modeling approaches were similarly utilized by Otero et al. [122], who delved into the creation of data-driven predictive models utilizing artificial intelligence to anticipate the occurrence of *Plasmopara viticola* and *Uncinula necator* in the viticultural regions of Southern Europe. Otero et al. employed a variety of machine and deep learning algorithms, including Logistic Regression, Decision Trees, Random Forest, Gradient Boosting, K-Nearest Neighbors, Naïve Bayes, Support Vector Machines, and Deep Neural Networks. Logistic Regression provides a probabilistic approach for binary outcomes, making it suitable for disease presence predictions. Decision Trees offer clear, interpretable decisions. Random Forest improves on Decision Trees by combining multiple trees to reduce overfitting. Gradient Boosting sequentially corrects errors, enhancing model performance. K-Nearest Neighbors classifies based on the majority vote of nearest data points, offering simplicity and effectiveness. Naïve Bayes, based on Bayes' theorem, excels in classification with an assumption of feature independence. Support Vector Machines efficiently handle high-dimensional data, optimizing margins between classes for clear decision boundaries. Notably, the models employed by Otero et al. achieved over 90% accuracy for infection risk and over 80% for treatment recommendations, highlighting the potential of AI in enhancing disease management strategies in vineyards across Southern Europe [122].

Collaboration and innovation in AI and cloud-based platforms are charting new paths in the monitoring and forecasting of plant diseases. The study published by Lavanya and Krishna [123] has developed a collaborative AI and cloud-based platform for plant disease identification, tracking, and forecasting. This innovative approach merges a mobile application with AI algorithms, providing real-time disease diagnostics and disease density mapping. This collaborative and technology-driven approach reflects a shift towards more integrated and interconnected systems for plant disease management, akin to the initiatives by Otero et al. [122] and Zen et al. [124]. The study by Zen et al. focuses on developing an AI-based mobile application for detecting plant diseases with high accuracy, utilizing CNN and RNN with TensorFlow.js. TensorFlow.js is an open-source library developed by Google for machine learning in JavaScript. It enables the training and deployment of machine learning models directly in the browser or on Node.js. TensorFlow.js provides a flexible and efficient platform for building and executing machine learning algorithms on web-based applications, allowing for interactive and real-time applications of AI technologies without the need for backend servers. The application in this study was tested on tomato plant diseases, achieving prediction accuracies of 100% for early blight, 90% for bacterial spot, and 100% for both healthy and late blight conditions. This research showcases the application's capability to recommend treatment options based on image analysis, offering a significant tool for farmers to identify and manage plant diseases effectively [122–124].

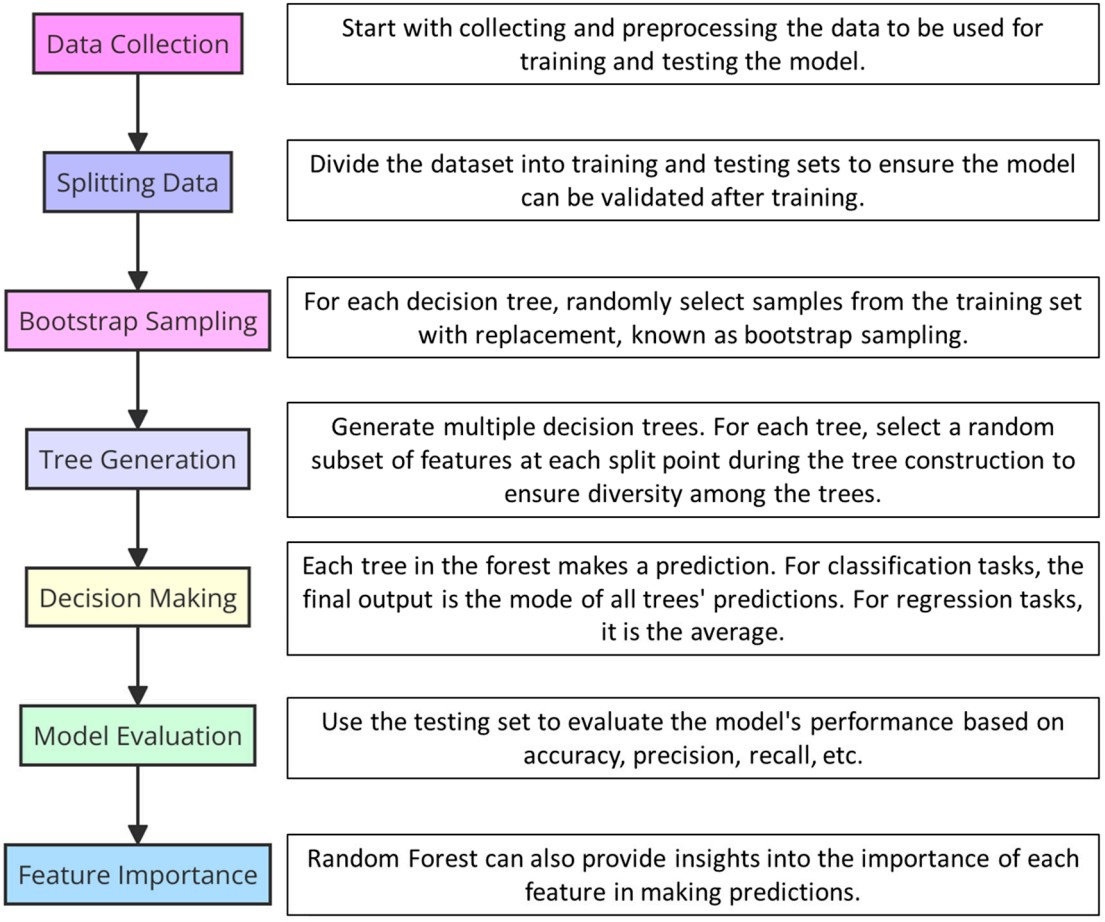

**Figure 4.** Graph diagram illustrating how Random Forest system works.

During 2023, in the realm of early disease detection and prediction in plants, technological advancements, particularly in Enhanced Data rates for GSM Evolution (EDGE) and deep learning, have played a pivotal role. Marco-Detchart et al. [125] focuses on the development of a robust, multi-sensor consensus approach for plant disease detection using the Choquet Integral. The Choquet Integral is a mathematical concept used in decision-making and information aggregation integrated in an Edge-AI device. Edge-AI refers to the deployment of AI applications directly on devices located at the "edge" of the network, rather than relying on centralized cloud services. This means that computations are performed close to where data are generated, such as in smartphones, surveillance cameras, or IoT devices, which allows for faster processing times, reduced bandwidth costs, and improved data privacy. This device was designed to improve disease classification by capturing multiple images of plant leaves and applying data fusion techniques. The system demonstrated increased robustness in classification responses to potential plant diseases, leveraging deep learning models for better analysis and classification. This innovatively implemented a multi-sensor consensus approach for plant disease detection, monitoring and prediction demonstrating efficacy surpassing traditional single-camera setups. Complementing this, Ojo and Zahid [126] have refined deep learning classifiers for plant disease detection by adeptly applying image preprocessing techniques and addressing class imbalance issues. Ojo and Zahid focused on enhancing deep learning classifiers for plant disease detection by addressing data imbalances and applying image preprocessing techniques. They used techniques like Adaptive Histogram Equalization (AHE), Contrast Limited Adaptive Histogram Equalization (CLAHE), and image sharpening to improve image quality. They tackled class imbalance with methods like Synthetic Minority Oversampling Technique (SMOTE), Major-to-Minor Translation (M2m), and Generative Adversarial Networks (GANs). These

studies collectively highlight the critical role of cutting-edge technology in efficient disease management and resource optimization in agricultural sectors [125,126].

The study by Vardhan et al. [127] takes an innovative approach to plant disease detection and monitoring using drone-captured imagery. They developed a comprehensive database from online sources, categorizing various plant species and diseases for analysis. This database was crucial for testing the accuracy and reliability of their CNN-based model. Their methodology emphasizes the use of CNNs due to their effectiveness in complex categorization and detection challenges, especially under varied imaging conditions. Additionally, they introduced a prototype drone equipped with a high-resolution camera for live field monitoring, showcasing the practical application of their research in real agricultural settings. This integration of advanced imaging techniques and AI algorithms represents a significant step forward in agricultural technology, offering a more efficient and accurate method for plant health assessment and disease management. In relation, the study published by Dagwale et al. [128] showed the YOLOv5 model, an advanced neural network architecture for real-time object detection, to accurately predict leaf species and diseases across various plant types using the PlantDoc dataset. YOLOv5 (You Only Look Once version 5) is designed for rapid image processing, identifying, and classifying multiple objects simultaneously with high precision. This integration showcases the potential of leveraging cutting-edge AI technologies like YOLOv5 to enhance disease detection accuracy in agriculture, marking a significant advancement in plant pathology diagnostics and disease spread and monitoring [128].

It is notably how neural networks, especially convolutional neural networks (CNNs), are emerging as the predominant technique for classifying plant diseases, thanks to their inherent flexibility and automatic feature extraction capabilities [124,127]. The developments shown in this subsection mark a milestone at the confluence of advanced technology and agronomy, heralding a new era in plant disease management. The fusion of machine learning techniques with cloud-based collaborative platforms is redefining the approach of farmers and scientists to plant disease challenges. These advancements not only enhance accuracy in disease detection and management but also facilitate a prompter and effective response, crucial for global sustainability and food security.

*5.3. Comprehensive Evaluation and Prospects of AI Technologies in Phytopathological Applications*

In the rapidly evolving domain of AI-assisted plant disease management, the integration of pretrained models, such as ResNet-18 and ResNet-50, marks a significant leap towards refining disease detection and diagnostic accuracy. These models, part of the Residual Networks (ResNets) introduced to mitigate the vanishing gradient problem in deep convolutional neural networks (CNNs), incorporate "shortcut connections" that allow gradients to flow through the network without undergoing non-linear transformations, thereby facilitating the training of much deeper networks. ResNet-50, a 50-layer CNN comprising 48 convolutional layers, one MaxPool layer, and one average pool layer, employs a "bottleneck" design in each residual block to reduce the number of parameters and accelerate layer training. This bottleneck design, featuring a stack of three layers instead of two, utilizes $1 \times 1$ convolutions to compress and then expand the number of channels, significantly lowering computational complexity while maintaining or enhancing model performance [129].

Originally trained on the expansive ImageNet dataset, these models exhibit exceptional prowess in feature recognition, offering tailored solutions for the nuanced challenges of phytopathology. Their capability to discern complex image characteristics with remarkable precision positions them as indispensable tools for identifying plant pathologies, often surpassing traditional visual inspection methods with accuracies ranging between 95% and 97%. Furthermore, the application of these pretrained models extends beyond disease identification to encompass broader spatial analyses, as evidenced by their deployment within the ArcGIS ecosystem for tasks like land cover classification and aerial feature

extraction, underscoring their potential to revolutionize the monitoring and management of plant health on a large scale [130].

The comparative analysis of AI models, including CNNs, YOLOv5, and MobileNet, illuminates the diverse applicability and efficacy of these technologies in phytopathology. Each model, with its unique strengths (be it CNNs for their image processing capabilities, YOLOv5 for its rapid processing speed facilitating timely interventions, or MobileNet for offering an efficient solution on low-power devices), advances our capacity to manage plant disease spread through predictive analyses that integrate environmental and symptomatic data. This synthesis not only augments diagnostic precision but also enhances proactive disease management strategies. Nevertheless, the practical application of these models encounters challenges such as the need for image preprocessing and handling unbalanced datasets, propelling the pursuit of technological innovations, especially in the realm of Edge-AI devices. These advancements promise a transformative impact on disease monitoring, enabling more accurate and accessible diagnostics.

As AI technologies continue to evolve, alongside breakthroughs in sensor technologies, we are ushered towards a new era of integrated and automated plant disease management. This journey is not without its hurdles, necessitating innovative approaches like transfer learning and the development of multisensorial detection systems to overcome current limitations. The ongoing exploration and refinement of AI models in phytopathology not only pave the way for future research directions but also highlight the pivotal role of AI in crafting sustainable, precision-based solutions for global agricultural challenges.

## 6. Applications of AI in Precision Agriculture and Management

The advent of artificial intelligence (AI) in precision agriculture marks a significant milestone in the evolution of precision farming, offering a promising avenue for enhancing yields while minimizing ecological impacts. The advantages of AI are not only related to phytopathology; groundbreaking developments in AI, such as advanced robotic weeders equipped with computer vision, have substantially reduced herbicide usage, exemplifying a move towards more sustainable farming practices [131]. Machine learning models that integrate weather, soil, and crop data have become increasingly sophisticated, aiding farmers in making well-informed decisions about irrigation, fertilization, and harvesting [132]. The democratization of AI through affordable solutions is further expanding the accessibility and effectiveness of precision farming [133].

The role of AI in agriculture, as outlined by Buja et al. [134], emphasizes the importance of early detection and rapid, accurate diagnostics for managing phytopathological challenges. This progress, marked by the application of nanotechnologies and the integration of the Internet of Things (IoT), is revolutionizing preventive strategies in combatting phytopathogens and precision agriculture. Liakos et al. [15] provide a comprehensive review of machine learning applications in agriculture, demonstrating how AI, combined with sensor data, is transforming farm management systems into real-time, intelligent platforms. These platforms offer insightful recommendations, significantly aiding in farmer decision-making across various aspects of agriculture, including crop, livestock, water, and soil management.

Kumar et al. [135] introduced DeepMC, a deep learning-based microclimate prediction framework utilizing IoT data, which exemplifies the potential of AI in enhancing precision agriculture. DeepMC's innovative approach to predicting a range of climatic parameters, including soil moisture, humidity, and temperature, offers accurate forecasts crucial for agricultural decision-making. The integration of AI in image processing has made significant contributions to precision agriculture. Studies by G S. and Rajamohan [136] and Sasikala D. and Sharma K. [137] demonstrate how AI-driven image processing technologies improve crop monitoring and management, further bolstering the efficiency and accuracy of agricultural practices.

Furthermore, the work of Joseph R.B. et al. [138] and Arokia Raj V.H. and Xavier de Carvalho C. [139] highlights the integration of AI in agricultural automation and agromete-

orology, respectively. These studies underscore AI's potential in enhancing the efficiency of agricultural products and in offering model-based decision support systems that unite AI with precision agriculture.

Lastly, Williams et al. [140] developed the AI2Farm model, a machine learning-based approach that analyzes the impact of global and domestic events on agricultural production, consumption, and pricing. This model represents a significant advancement in precision agriculture by providing farmers with tools to adapt to both conventional and unconventional challenges in agriculture.

In summary, the integration of AI into precision agriculture and management marks a transformative shift in modern farming encompassing sustainable practices, advanced diagnostics, data-driven decisions, and innovative technologies. These developments are crucial to meet escalating food demands while maintaining ecological balance. Concurrently, manifold AI applications in detection, forecasting, precision management, and monitoring are transforming phytopathology. As these techniques mature, AI-enabled tools promise to strengthen global food security and agricultural sustainability amidst evolving plant disease challenges. Taken together, the advent of precision agriculture powered by AI constitutes a strategic innovation pathway for next-generation phytopathology and plant protection practices worldwide. With prudent and ethical implementation, data-driven smart farming technologies can enable the sustainable intensification of crop productivity to feed rising populations in the face of climate change and agricultural pressures.

## 7. Integration Challenges and Ethical Considerations

### 7.1. Technical Barriers to AI Implementation

While artificial intelligence promises transformative phytopathology innovations, prudent precautions are necessary for its successful integration into agricultural systems. Technical barriers persist in developing robust, reliable AI solutions for real-world plant disease environments [47]. A key limitation of many current machine learning models is their narrow focus on specific crops, pathogens, and controlled settings [141]. Algorithms trained on limited datasets often fail to generalize across diverse agricultural contexts. The myriad variations in crop cultivars, growth stages, climates, soil conditions, and pathogen strains pose challenges in creating AI tools with sufficient flexibility for in situ usage [13,142].

Progress is also impeded by a lack of coordination across data collection efforts and an unwillingness to openly share datasets between research groups and private entities. Most available plant disease datasets remain relatively small-scale and sparse [14]. Such fragmented data restrict the training and performance scope of AI systems. While emerging sensor, imagery, and genomic technologies offer copious agricultural data streams, integrating such disparate formats for AI utilization is non-trivial and requires dedicated preprocessing pipelines [15].

Researchers have outlined frameworks to methodically address these technical barriers through good data practices and coordinated action [141,143]. Recommendations include collaborative open-access data platforms, standardized collection protocols, and emphasis on creating shareable datasets with diversity. Transfer learning methods that leverage models pre-trained on large natural image repositories are also being explored to improve generalization despite limited domain-specific agricultural data [144].

In additions, while AI holds the potential to drive sustainable agricultural practices, such as optimizing resource use and minimizing chemical inputs, it is also essential to consider the environmental footprint of the AI technology itself. This includes the carbon footprint associated with data centers powering AI applications and the environmental impact of manufacturing AI hardware. Sustainable AI in agriculture should strive for a balance where the ecological benefits of its application significantly outweigh its environmental costs [145].

### 7.2. Ethical Issues in AI-Driven Phytopathology

The integration of AI into phytopathology or precision agriculture also raises pressing ethical concerns regarding data privacy, accountability, labor impacts, and environmental sustainability that warrant scrutiny [146,147]. Critics caution that AI-enabled crop management regimes could reinforce unsustainable industrial farming at the cost of rural livelihoods, localized knowledge, and the food sovereignty of smallholder farmers [148,149]. Therefore, phytopathology AI systems must be designed through inclusive stakeholder participation, centering human needs and values.

There is apprehension surrounding the data privacy and consent procedures involved in collecting large agricultural datasets for training AI models, which could include farmer proprietary information alongside field images or soil data [36]. The onus is on researchers to implement ethical data management practices that protect farmer interests and anonymity. Moreover, the proprietary black-box nature of some commercial AI technologies obscures model biases and prevents oversight into decision-making rationales [150] (Ribeiro et al., 2016). Such opacity becomes ethically problematic for AI systems deployed in social realms including agriculture [151].

Broader concerns also exist around delegating data-driven farming fully to AI, potentially marginalizing rural communities and eroding farmers' autonomy, knowledge, and sense of place [148]. Hence, human-centered design considerations must shape responsible AI integration in phytopathology, serving to augment, not replace, agricultural expertise and intuition. Ongoing farmer education and upskilling initiatives are imperative to democratize AI access, allowing rural communities to reap the benefits equitably and partake in co-developing solutions attuned to local needs [43,152].

### 7.3. Regulatory Frameworks and Standards

Realizing ethical AI for agriculture requires establishing regulatory frameworks and technological standards guiding development and deployment [153]. At present, there is a lack of governance surrounding the creation, sales, and monitoring of AI phytopathology technologies. Policy interventions are required at national and global levels to regulate the quality control, risk assessments, and liability attribution of agricultural AI systems. Such oversight can mitigate dangers of hastily implemented tools with unreliable real-world performances or unexamined biases causing harm [154,155].

Global agreements are also needed to align technological approaches, architecture choices, data formats, curation protocols, and performance benchmarks across the emerging field of AI phytopathology [156]. Common technology standards will support collaboration, open data sharing, and interoperability, accelerating innovation. Furthermore, voluntary professional codes of ethics around topics such as model transparency, auditability, and farmer privacy could guide institutional research and industry product design [157]. Overall, multi-tiered governance combining binding regulations and soft-law ethics codes can steer agricultural AI progress along responsible trajectories.

In summary, realizing AI's promise in transforming 21st century phytopathology necessitates prudently navigating the accompanying integration barriers and ethical tensions. Only through holistic frameworks considering all stakeholder needs can AI technologies serve humanity in enabling sustainable plant disease management worldwide. The path forward lies in an inclusive and value-based co-development of agricultural AI tools, supported by emergent policy regimes governing these evolving technologies for societal benefit.

## 8. Conclusions

### 8.1. Summary of Key Findings

This comprehensive review highlights the immense potential of artificial intelligence to transform modern approaches in plant disease management and phytopathological research. Through an extensive analysis of the existing literature, manifold AI applications across major facets of phytopathology have been delineated.

Notable successes have been demonstrated in employing AI for automated disease detection and diagnosis using image recognition techniques like convolutional neural networks. Studies indicate that AI can identify plant diseases, often with 95–97% accuracy, exceeding human visual inspection. AI also shows an aptitude for data-driven disease spread forecasting, integrating weather, soil, and crop parameters to predict outbreak risks up to 3 weeks prior at 81–95% precision. This enables preemptive and targeted protection strategies minimizing pesticide usage.

Furthermore, AI is optimizing precision agriculture through site-specific interventions tailored to local conditions based on integrated crop data analysis. These holds promise to boost yields while protecting ecosystems. The AI monitoring of plant and environmental cues also facilitates pre-symptomatic disease alerts for early action. Ongoing research on explainable and transparent AI can mitigate issues surrounding model opacity.

Overall, real-world evidence affirms that AI-enabled tools can strengthen disease control, enhance crop resilience, and unlock novel phytopathological insights from increasingly complex agricultural data streams [158]. AI's self-improving and generalizable capabilities make it well suited to address evolving plant health challenges amidst climate change, globalization, and intensified farming systems.

## 8.2. Implications for the Future of Phytopathology

The integration of AI portends a paradigm shift in phytopathology and plant protection strategies worldwide. As algorithms become more robust and tailored for agricultural settings, AI's role is poised to expand from assisting tasks to autonomous in-field decision-making around disease management. With sufficient training data encompassing diverse cropping contexts, AI systems can attain the flexibility and adaptiveness required for broad deployment.

Advances in sensors, automation, and robotics will enable expansive data generation on crop status, disease progression, and environmental influences. AI's capacity to assimilate such big data and discern correlations can illuminate plant–microbe interactions, evolutionary dynamics, and epidemiology at unprecedented resolution. These insights promise to accelerate knowledge discovery and innovation in phytopathology, seeding 21st century breakthroughs.

Overall, the advent of data-driven smart farming powered by AI algorithms marks a historic juncture in tackling plant disease burdens. As phytopathology transitions into an interdisciplinary, technology-intensive science, AI will catalyze a strategic shift towards precision and sustainable agriculture. This new paradigm seeks ecological disease prevention over chemical controls, supporting global food security and environmental objectives.

## 8.3. Call to Action for Researchers and Stakeholders

Realizing the immense promise of AI in enabling next-generation phytopathology necessitates focused efforts by researchers worldwide alongside multi-stakeholder participation. Key priorities include assembling diverse open-access datasets, advancing collaborative models, strengthening farmer education, and developing supportive policies.

Researchers must coordinate shared protocols and create expansive training datasets encompassing various crops, cultivars, pathogens, and agricultural environments. This will improve AI model robustness, avoiding dataset limitations. Committing to open data access and developing regional repositories are critical to accelerated innovation. Advancing participatory models where farmers help co-design context-specific AI tools is vital to democratize benefits equitably. Capacity building to equip farming communities in adopting smart technologies responsibly is imperative. Policymakers must also implement updated regulations governing agricultural AI development and deployment for the public good.

In summary, the promising future of AI in plant disease management calls for collective action by stakeholders worldwide. Through ethically aligned, inclusive efforts that put farmers first, AI can help secure the productivity and sustainability of agricultural systems

globally in the face of rising pressures. This necessitates bridging disciplinary divides and steering agricultural AI progress along humanistic trajectories.

**Author Contributions:** All the authors collaborated in the research and review described in this article. Conceptualization, V.E.G.-R. and I.I.-B.; investigation, V.E.G.-R., I.I.-B., J.M.C., M.C. and C.G.; writing—original draft preparation, V.E.G.-R. and I.I.-B.; writing—review and editing, J.M.C., M.C., and C.G.; supervision, M.C. and C.G.; funding acquisition, J.M.C., M.C. and C.G. All authors have read and agreed to the published version of the manuscript.

**Funding:** This research was supported by grants from MCIN PID2021-122899O-B-C22 MCIN/AEI/ FEDER, EU) and the University of Cádiz through the "Programa de Fomento e Impulso de la actividad de Investigación y Transferencia de la Universidad de Cádiz".

**Data Availability Statement:** Not applicable.

**Conflicts of Interest:** The authors declare no conflicts of interest.

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
