# Peer review of "Artificial Intelligence: A Promising Tool for Application in Phytopathology"

_horticulturae, doi:10.3390/horticulturae10030197_

Round 1

Reviewer 1 Report

Comments and Suggestions for Authors

This paper reviews the application of artificial intelligence technology in the study of plant diseases, covering the current study and basic knowledge of artificial intelligence, the application of AI in plant disease research, and even its potential issues and development direction. The article still has some shortcomings as follows:

(1)Overall, the main focus of this article should be on the application of AI in detecting plant diseases, as indicated by the article title. About "Phytopathology" and artificial intelligence technology, the content is supposed to be reduced.

(2)From line 172 to line 355, a significant amount of space is devoted to introducing the fundamental knowledge of artificial intelligence. However, this information has weak relevance to the main focus of this article, which should be on plant diseases.

(3)As a review article, this article lacks performance evaluation comparisons for different artificial intelligence methods in plant disease detection, advancements in modeling, etc.

(4)In Section 5, "Applications of AI in Phytopathology", it is noted that "5.3 Precision Agriculture and Management" does not pertain to the topic of "Phytopathology" discussed in the article. The definition of "Phytopathology" in the text is "...scientific discipline...the study of plant diseases", as stated in line 70.

(5) It is suggested to add some images to represent the principles of various artificial intelligence methods visually.

Author Response

Dear reviewer 1:

Many thanks for your deep analyses of our manuscript and for your constructive suggestions. We have revised in deep the manuscript and we have follow all your recommendation of improvements. Next, you will find our responses.

Shortcomings

Response

1

Overall, the main focus of this article should be on the application of AI in detecting plant diseases, as indicated by the article title. About "Phytopathology" and artificial intelligence technology, the content is supposed to be reduced

We have extensively expanded section 5, with greater depth in the analysis of each work examined and with an increased amount of explanations about the methods used and their results. Additionally, we have included a new subsection 5.3 discussing the aforementioned methods and comparing them with each other.

2

From line 172 to line 355, a significant amount of space is devoted to introducing the fundamental knowledge of artificial intelligence. However, this information has weak relevance to the main focus of this article, which should be on plant diseases.

We agree with the reviewer that the section dedicated to the fundamentals of AI is somewhat lengthy. We believe that this information serves as a good illustration and introduction for readers not familiar with these fundamentals. Nevertheless, following the recommendations made, we have shortened its length from about 1600 words to just 1000 words, resulting in a reduction of more than 35%.

3

As a review article, this article lacks performance evaluation comparisons for different artificial intelligence methods in plant disease detection, advancements in modeling, etc.

we have included a new subsection 5.3 discussing the aforementioned methods and comparing them with each other.

4

In Section 5, "Applications of AI in Phytopathology", it is noted that "5.3 Precision Agriculture and Management" does not pertain to the topic of "Phytopathology" discussed in the article. The definition of "Phytopathology" in the text is "...scientific discipline...the study of plant diseases", as stated in line 70.

Although the Precision Agriculture and Management section is not directly related to phytopathology as indicated by the reviewer, we consider the information developed and reviewed within it to be relevant and related. Therefore, following the criteria set out by reviewer 1, this subsection has been extracted from section 5 and a new section titled 6. Applications of AI in Precision Agriculture and Management has been created, subsequently renumbering the rest of the sections according to the new layout.

5

It is suggested to add some images to represent the principles of various artificial intelligence methods visually

Two new figures have been generated to illustrate the article a bit: Figure 3. Graph diagram illustrating the operation of Siamese Convolutional Neural Networks (SNNs) for detecting plant diseases from an image dataset; and Figure 4. Graph diagram illustrating how the Random Forest system works.

We are immensely grateful for the reviewer's comments, as they have significantly contributed to the improvement of our manuscript. We sincerely hope that this revised version meets your approval and that it is suitable for publication in the journal.

Yours sincerely,

Reviewer 2 Report

Comments and Suggestions for Authors

1. Incorrect subheader format: 4.3 Relevance of AI in Various Fields, 4.3.5 Space Exploration: Autonomous Exploration and Data Analysis,

2. From my perspective, there is a disproportionate allocation of content with approximately 9.5 pages dedicated to general information about AI as opposed to 6.5 pages about Phytopathology

3. The article evidently appears to have been composed using ChatGPT

4. It may be beneficial to consider reviewing the following papers about

a) Vision Transformers: A. Tabbakh and S. S. Barpanda, "A Deep Features Extraction Model Based on the Transfer Learning Model and Vision Transformer “TLMViT” for Plant Disease Classification," in IEEE Access, vol. 11, pp. 45377-45392, 2023, doi: 10.1109/ACCESS.2023.3273317. keywords: {Feature extraction;Plant diseases;Support vector machines;Transfer learning;Machine learning;Lesions;Image color analysis;Deep learning;Image processing;Plant disease;image processing;deep learning;transfer learning;vision transformer}, B. Detection and Classification of Plant Stress Using Hybrid Deep Convolution Neural Networks: A Multi-Scale Vision Transformer Approach. https://doi.org/10.18280/ts.400625

b) Siamese network:

https://link.springer.com/chapter/10.1007/978-981-19-2535-1_3 https://doi.org/10.1007/978-981-19-2535-1_3

5. In Table 1, there is an absence of traditional machine learning approaches such as:

a. Multilayer Perceptron (MLP)

b. Stochastic Gradient Descent (SGD)

c. Random Forest

d. ElasticNet/Lasso/Ridge Regression

e. Decision Tree/XGBoost

For example: https://doi.org/10.3390/agriculture12122089

6. There is a lack of discussion on existing pretrained models that could be applied in Phytopathology.

7. Is there any application of Explainable Artificial Intelligence (XAI) in Phytopathology? It would be worthwhile to engage in a discussion on this topic. For instance: https://www.diva-portal.org/smash/get/diva2:1593851/FULLTEXT01.pdf

Comments on the Quality of English Language

From my point of view english level is at the good level. The paper evidently appears to have been composed using ChatGPT.

Author Response

Dear reviewer 2:

Many thanks for your deep analyses of our manuscript and for your constructive suggestions. We have revised in deep the manuscript and we have follow all your recommendation of improvements. Next, you will find our responses.

Shortcomings

Response

1

Incorrect subheader format: 4.3 Relevance of AI in Various Fields, 4.3.5 Space Exploration: Autonomous Exploration and Data Analysis,

This mistakes have been corrected.

2

From my perspective, there is a disproportionate allocation of content with approximately 9.5 pages dedicated to general information about AI as opposed to 6.5 pages about Phytopathology

We agree with the reviewer that the section dedicated to the fundamentals of AI is somewhat lengthy. We believe that this information serves as a good illustration and introduction for readers not familiar with these fundamentals. Nevertheless, following the recommendations made, we have shortened its length from about 1600 words to just 1000 words, resulting in a reduction of more than 35%.

Additionally, we have extensively expanded section 5, with greater depth in the analysis of each work examined and with an increased amount of explanations about the methods used and their results. A new subsection 5.3 discussing the aforementioned methods and comparing them with each other was included.

Two new figures have been generated to illustrate the article a bit: Figure 3. Graph diagram illustrating the operation of Siamese Convolutional Neural Networks (SNNs) for detecting plant diseases from an image dataset; and Figure 4. Graph diagram illustrating how the Random Forest system works.

3

The article evidently appears to have been composed using ChatGPT

Some computer applications were used to translate some expressions and phrases from Spanish to English, including AI tools. We have conducted a thorough review of the manuscript to try to avoid some repeated words or phrases that were erroneously included as a result of these automated correction processes. Thank you for pointing this out so that the current version can be of higher quality.

This thorough review, which includes 157 references, many of them very recent, has been crafted and elaborated with great care and analysis, and based on our knowledge and scientific experience in writing original articles, reviews, and book chapters by the team of authors listed in the review.

4

It may be beneficial to consider reviewing the following papers about

We appreciate the reviewer's recommendations for potential works. We have consulted the references and found it highly interesting to include the themes of vision transformers and siamese networks. These works are extensively explained in section 5.1 between lines 394 and 469. We have even included a figure 3 about the SNNs workflow.

5

In Table 1, there is an absence of traditional machine learning approaches such as:

a. Multilayer Perceptron (MLP)

b. Stochastic Gradient Descent (SGD)

c. Random Forest

d. ElasticNet/Lasso/Ridge Regression

e. Decision Tree/XGBoost

We found the suggestion made by the reviewer very interesting and in line with that, we have updated and expanded the information in table 1 with the proposed models.

6

There is a lack of discussion on existing pretrained models that could be applied in Phytopathology.

A new subsection 5.3 discussing the aforementioned methods and comparing them with each other was included. This subsection include mention to pretrained models.

7

Is there any application of Explainable Artificial Intelligence (XAI) in Phytopathology? It would be worthwhile to engage in a discussion on this topic

The studies on Explainable Artificial Intelligence (XAI) are both intriguing and promising, yet there are currently no published studies on its application specifically within the field of phytopathology. However, XAI has seen application in various other domains, such as healthcare management, industrial asset prognostics, and electronic health record analysis, showcasing its broad potential and utility.

We are immensely grateful for the reviewer's comments, as they have significantly contributed to the improvement of our manuscript. We sincerely hope that this revised version meets your approval and that it is suitable for publication in the journal.

Round 2

Reviewer 1 Report

Comments and Suggestions for Authors

Thanks to the author for the revisions and replies, I have no further comments.

Reviewer 2 Report

Comments and Suggestions for Authors

All comments have been appropriately addressed